# A Critical Overview of Targeted Therapies for Vestibular Schwannoma

**DOI:** 10.3390/ijms23105462

**Published:** 2022-05-13

**Authors:** Ryota Tamura, Masahiro Toda

**Affiliations:** Department of Neurosurgery, Keio University School of Medicine, 35 Shinanomachi, Shinjuku-ku, Tokyo 160-8582, Japan; todam@keio.jp

**Keywords:** schwannoma, NF2, bevacizumab, VEGF, SH3PXD2A-HTRA1 fusion, molecular targeted therapy

## Abstract

Vestibular schwannoma (VS) is a benign tumor that originates from Schwann cells in the vestibular component. Surgical treatment for VS has gradually declined over the past few decades, especially for small tumors. Gamma knife radiosurgery has become an accepted treatment for VS, with a high rate of tumor control. For neurofibromatosis type 2 (NF2)-associated VS resistant to radiotherapy, vascular endothelial growth factor (VEGF)-A/VEGF receptor (VEGFR)-targeted therapy (e.g., bevacizumab) may become the first-line therapy. Recently, a clinical trial using a VEGFR1/2 peptide vaccine was also conducted in patients with progressive NF2-associated schwannomas, which was the first immunotherapeutic approach for NF2 patients. Targeted therapies for the gene product of SH3PXD2A-HTRA1 fusion may be effective for sporadic VS. Several protein kinase inhibitors could be supportive to prevent tumor progression because merlin inhibits signaling by tyrosine receptor kinases and the activation of downstream pathways, including the Ras/Raf/MEK/ERK and PI3K/Akt/mTORC1 pathways. Tumor-microenvironment-targeted therapy may be supportive for the mainstays of management. The tumor-associated macrophage is the major component of immunosuppressive cells in schwannomas. Here, we present a critical overview of targeted therapies for VS. Multimodal therapy is required to manage patients with refractory VS.

## 1. Introduction

Schwann cells originate from neural crest cells, which migrate with growing neurites during nerve development. Schwann cells, which form the myelin sheath of an axon, support neuronal function and regeneration [1].

Schwannoma (Sch) is one of the common benign intracranial tumors with an incidence of 1 per 100,000 [2]. Sch often presents between the ages of 40 and 60 years [2]. Among these cases, 80–90% originate from the vestibular nerve. About 5–10% of vestibular Schs (VSs) are observed as bilateral in neurofibromatosis 2 (NF2) patients. A total of 95% of NF2 patients show bilateral VSs [3]. About 60% of unilateral VSs and 90% of bilateral VSs show NF2 gene mutation and the dysfunction of its transcription product, moesin–ezrin–radixin-like (merlin) protein [4].

Currently, the mainstays of management are observation, surgery, and radiosurgery. Surgery with facial and auditory monitoring remains the only curative treatment for growing VSs of all sizes. Stereotactic radiosurgery is considered as a widely accepted treatment option for small-sized VSs. For larger tumors, combined treatment strategies are mostly recommended. In particular, gamma knife radiosurgery (GKRS) has become an accepted treatment for VS [5]. However, additional treatment is needed for some refractory cases. Tumor volume ≥15 cm^3^ is a significant factor predicting poor tumor control following GKRS [6]. There is no approved medical therapy for VS. For refractory VS with high risks of surgical treatment or GKRS, medical therapies that can slow tumor growth are urgently needed. Here, we review the molecular biology and its relevance to treatment for VS.

## 2. NF2 Gene

NF2 is an autosomal-dominant disease caused by a biallelic loss of the NF2 gene on chromosome 22. Although 50% of NF2 patients have an affected parent with the disease, the remaining 50% have de novo gene mutations [7].

Although 60% of patients with de novo NF2 show mosaic NF2, the actual diagnostic rate of this condition remains low at 20% because of the difficulties in detecting NF2 variants with a low variant allele frequency [8]. Teranishi et al. improved the diagnostic rate of mosaic NF2 using targeted deep sequencing of DNA. The mosaic NF2 phenotype was found to be different from that in the NF2 germline variant in terms of tumor growth and hearing outcome [8].

Differentiated Schwann cells become quiescent because merlin regulates this contact-dependent inhibition of proliferation. Merlin plays a significant role in regulating the actin cytoskeleton, adhesion junction formation, and cell proliferation [9]. Merlin can regulate multiple tumorigenic pathways, including retrovirus-associated DNA sequences (Ras)/rapidly accelerated fibrosarcoma (Raf)/mitogen extracellular signal-regulated kinase (MEK)/extracellular-signal-regulated kinases (ERK), and mammalian target of rapamycin complex 1 (mTORC1)/phosphoinositide 3-kinase (PI3K)/Akt [10,11].

## 3. SH3PXD2A-HTRA1 Fusion

In 2016, alternative tumorigenic mechanisms were proposed, including a recurrent in-frame fusion transcript of the HTRA1 and SH3PXD2A genes. The gene product of SH3PXD2A-HTRA1 fusion promotes proliferation and invasion. In a previous study, the frequency of this fusion gene was investigated [12]. The fusion gene SH3PXD2A-HTRA1, activating the MAPK pathway, has been associated with 10% of sporadic Schs. Agnihotri et al. suggested that SH3PXD2A-HTRA1 fusion promoted tumorigenesis and sensitivity to an MEK-ERK inhibitor [12]. Even though SH3PXD2A-HTRA1 fusion has been shown to be a driver of tumorigenesis, the fusion transcript was extremely rare in Norwegian sporadic VS patients [13]. Further investigation is warranted to elucidate the importance of this fusion gene.

## 4. Protein-Kinase-Related Pathway

### 4.1. VEGF-A/VEGFRs

The vascular endothelial growth factor receptor (VEGFR) family mainly includes VEGFR-1 (Flt-1), VEGFR-2 (Flk-1/KDR), and VEGFR-3 (Flt-4), which are important regulators of physiological and pathological angiogeneses [14]. Merlin deletion leads to the downregulation of the protein semaphorin 3F, which inhibits VEGF-mediated angiogenesis [15]. A previous study has shown that the concentrations of VEGF-A and VEGFR-1 are related to the growth rate of VS [16].

Tumor shrinkage and hearing improvement have been identified after the administration of bevacizumab (anti-VEGF-A antibody) in about 41% and 20% of progressive VSs in NF2 patients, respectively [17]. Bevacizumab may be considered as first-line medical therapy for rapidly growing VS. In a recent meta-analysis, the median treatment duration was 16 months [18,19]. Recently, the first phase III randomized clinical trial using bevacizumab was conducted in Japan [20,21]. Furthermore, progressive sporadic VS also exhibited significant tumor shrinkage after bevacizumab administration [22].

However, some aspects of bevacizumab treatment are problematic, such as the need for frequent parenteral administration, side effects, apparent drug resistance, and rebound tumor progression [23]. In the majority of published case series of bevacizumab usage for VS, their conclusions on efficacy were based on relatively short follow-ups. Long-term follow-up studies using a large number of patients are warranted. A clinical trial using a VEGFR-1/2 peptide vaccine was also conducted in patients with progressive NF2-derived Schs, showing hearing improvement and tumor volume reduction. Memory cytotoxic T lymphocytes have the possibility to persist in the long-term [24]. This was the first immunotherapeutic approach for NF2 patients.

### 4.2. ErbB

The ErbB family’s cell membrane receptors include the epidermal growth factor receptor (EGFR) (HER1/ErbB-1), HER2 (neu/ErbB-2), HER3 (ErbB-3), and HER4 (ErbB-4). MAPK/ERK and PI3K/Akt signaling pathways are considerably downstream of ErbB-2 activation [25]. ErbB receptors were activated in both sporadic and NF2-related VSs, and EGFR expression levels correlated with Sch size [26]. Furthermore, EGF was upregulated in NF2-related VS but not in sporadic VS, suggesting that an EGFR inhibitor might have efficacy in NF2 patients [27,28].

The predominant ErbB receptor dimerization patterns in VS are EGFR and ErbB2 heterodimers [29]. Trastuzumab, a humanized anti-ErbB2 monoclonal antibody, could significantly reduce tumor growth; however, this antibody did not induce significant cell death in VS xenografts [29].

Lapatinib is a potent and reversible tyrosine kinase inhibitor, showing a dual inhibitory effect on the EGF activation of EGFR/ErbB2 [30]. A phase II clinical trial showed that lapatinib has minor toxicity and the minor effects of reducing tumor volume and improving hearing in NF2-related progressive VS [30]. This treatment failure was due to the ErbB3 upregulation caused by the inhibition of ErbB2. Erlotinib is a reversible, small-molecule EGFR-specific tyrosine kinase inhibitor [30]. However, erlotinib was ineffective in NF2-related VSs for tumor shrinkage and improving hearing outcome. Bevacizumab has shown better benefits in the treatment of NF2 patients compared with lapatinib and erlotinib [31].

### 4.3. PDGFR

Platelet-derived growth factor (PDGF) regulated the migration of mesenchymal stem cells via PI3K signaling [32]. The PDGF receptor (PDGFR) family includes PDGFR-α, PDGFR-β, colony-stimulating factor1 receptor (CSF1-R), fetal liver kinase-2 (Flk-2), and c-kit [32]. Compared with normal nerves, the expressions of c-kit, PDGFR-α, and PDGFR-β are increased in sporadic and NF2-related VSs [33]. Imatinib mesylate (STI571) is an inhibitor of the BCR-ABL fusion kinase for chronic myelogenous leukemia [34]. Imatinib mesylate inhibits the activation of c-KIT, PDGFR-α, and PDGFR-β and their downstream signaling pathways, leading to increased apoptosis in the immortalized NF2-null VS cell line. Moreover, imatinib has an inhibitory effect for angiogenesis in both sporadic and NF2-related VSs [35].

Nilotinib (Bcr-Abl tyrosine kinase inhibitor) is 10–30-fold more potent than imatinib in inhibiting Bcr-Abl tyrosine kinase activity and proliferation [36]. Nilotinib also inhibited cell proliferation more effectively compared with imatinib in Sch cell lines. Anti-tumor effects were related to the inhibition of PDGFR-α and PDGFR-β, as well as their downstream signaling mediators, Akt and mTOR [36].

Ponatinib inhibits SRC, fibroblast growth factor receptor (FGFR), PDGFR, and VEGFR1–3, stimulating a robust G1 cell cycle arrest of merlin-deficient human Schwann cells [37]. However, in the clinical setting, targeting PDGF/PDGFR signaling did not show significant benefits in the treatment of NF2 patients.

### 4.4. HGFR

The hepatocyte growth factor receptor (HGFR), known as c-mesenchymal–epithelial transition (c-MET), is a glycosylated receptor tyrosine kinase and plays a role in driving tumorigenesis [38,39]. The activation of the HGF/c-MET pathway in sporadic VS can promote the inflammation network and cancer progression [40]. This pathway can also protect cells from apoptosis induced by chemotherapy or radiotherapy through PI3K/Akt signaling [41].

Therefore, crizotinib (a c-MET and anaplastic lymphoma kinase inhibitor) can enhance the radiation-induced DNA damage of NF2-related Sch cells, enhancing radio sensitivity. This effect leads to a reduction in radiation dose and protects hearing [42]. A phase II clinical trial using crizotinib for NF2 and progressive sporadic VSs in children and adults is ongoing (NCT04283669). The simultaneous use of the c-MET inhibitor “cabozantinib” and the Src inhibitor “saracatinib” can reduce the viability of human VS cells with he NF2 mutation, which is more effective than using either inhibitor alone [43].

There is a crosstalk between c-MET and VEGF-A in VSs. Sonam et al. found that c-MET and VEGF-A protein levels decreased using c-MET-targeted siRNA, while VEGF-A- targeted siRNA reduced c-MET expression. The combined inhibition of VEGF-A and c-MET may be an effective therapy [40].

### 4.5. PI3K/Akt/mTOR

PI3K/Akt/mTOR signaling contributes to a variety of processes that are critical in anabolic reactions and cell growth and survival. PI3K/Akt/mTOR signaling is elevated in VS [44]. Therefore, the PI3K/Akt pathway is also an attractive treatment target for VS [44].

OSU-03012 is an ATP-competitive inhibitor of PAK activity and suppresses the phosphorylation of Akt, which inhibits VS cell growth and promotes apoptosis [45]. Additionally, OSU-HDAC42 (AR-42), a novel phenylbutyrate-derived histone deacetylase inhibitors, can inhibit the downstream Akt expression of PI3K through protein phosphatase-1-mediated Akt dephosphorylation, showing the effect of G2 cell cycle arrest and cell apoptosis in a VS animal model [46].

mTOR is a downstream signal of the PI3K/Akt pathway [47]. A previous study has shown that an mTORC1 inhibitor (rapamycin) can inhibit the growth of merlin-deficient tumors in vivo. Rapamycin can lead to tumor shrinkage in NF2 patients with growing VSs [48].

Everolimus (RAD001), a derivative of rapamycin, can inhibit mTORC1 and reduce tumor angiogenesis. Although a phase II study has shown that everolimus is ineffective in progressive NF2-related VS patients [49], another study has shown that everolimus reduced the tumor volume in 55.6% of patients with NF2-related VS [50,51]. The effect of everolimus is still debatable.

## 5. Cytokines and Chemokines

C-X-C motif chemokine ligand 12 (CXCL12) binds to C-X-C chemokine receptor type 4 (CXCR4). The CXCL12/CXCR4 axis plays a pivotal role in tumor development, survival, angiogenesis, metastasis, and the tumor microenvironment. In addition, this chemokine axis promotes chemoresistance in cancer therapy. CXCR4 is also considered to be correlated with the tumorigenesis and functional disturbance of sporadic and NF2-related VSs [52]. CXCR4-directed positron emission tomography/computed tomography imaging with radiolabeled CXCR4-targeted ligand [68Ga]-Pentixafor was used to evaluate CXCR4 expression in VS patients [53]. These results provide a possibility for the use of Plerixafor (AMD3100) as a CXCR4-targeting drug [52,53].

Multiple cytokines and chemokines, including CXCL12, CXCL16, interleukin (IL)-1β, IL-6, IL-34, macrophage colony-stimulating factor (M-CSF), and tumor necrosis factor-α (TNF-α), are also associated with tumor progression and hearing disturbance [54].

In addition to the direct compression of auditory nerve fibers by tumors, in cases of NF2-associated deafness, detrimental paracrine substances, such as proinflammatory cytokines from tumors, have been proposed as a mechanism of cochlear hearing loss [55]. A novel therapeutic strategy targeting cytokines and chemokines may support other treatment strategies.

## 6. Tumor Microenvironment

Sch consists of different cell types, including tumorigenic Schwann cells, axons, macrophages, T cells, fibroblasts, blood vessels, and an extracellular matrix. The tumor microenvironment plays a relevant role in the development and progression of Sch. There are few studies regarding the tumor microenvironment in Sch [56,57].

Fast-growing VSs expressed high M-CSF and IL-34 levels that could regulate the chemotaxis of tumor-associated macrophages (TAMs). TAMs produce growth factors and anti-inflammatory cytokines to suppress the host immune response, resulting in tumor progression. VEGF in the hypoxic tumor microenvironment is a key factor for transitioning from the M1 to the M2 macrophage phenotype [58]. A greater TAM infiltration was found in growing sporadic VSs compared with non-growing sporadic VSs [54,59,60].

Programmed death-1 (PD-1) is expressed on CD8+T cells. Programmed death-ligand 1 (PD-L1) is expressed on tumor cells in numerous malignant tumors and binds to PD-1 to negatively regulate the immune response of CD8+T cells [58]. In 11 NF2-associated Schs, both high levels of programmed death-ligand 1 (PD-L1) expression and the presence of TAMs and T lymphocytes were identified in nearly all specimens [61]. In another study of 44 sporadic Schs, an increased presence of TAMs and an elevated PD-L1 expression were significantly associated with tumor progression [62].

Regulatory T cells (Tregs) (CD4 + CD25 + Foxp3+) play an active and significant role in the progression of tumors, and they play an important role in suppressing tumor-specific immunity [58]. In NF2 patients, the number of Foxp3-positive cells in Sch with a progressive course was significantly higher than in those without a progressive course, suggesting that growth may be associated with Foxp3-positive Tregs [24,63].

A previous study investigated the hypoxic tumor microenvironment of patients with NF2 Sch. Hypoxia was important for the shorter progression-free survival of NF2 Sch [59]. An immunotherapy that specifically targets the tumor microenvironment may emerge as a new class of Sch therapeutics.

## 7. Inflammation and Stress Reaction

### 7.1. COX2

The expression of cyclooxygenase 2 (COX-2) is associated with sporadic and NF2-related VS proliferation. Mutations in the NF2 gene can activate the Hippo pathway, in which YAP can promote the transcription of COX-2 for prostaglandin production. Prostaglandin E2 (PGE2) catalyzed by COX-2 has multiple roles in cell proliferation, apoptosis, angiogenesis, inflammation, and immune monitoring. COX-2 inhibitors may have the potential to inhibit the growth of VS [64,65].

A negative correlation between aspirin users and sporadic VS growth has been demonstrated [66,67]. In addition to inhibiting COX-2, aspirin can also suppress the activated NF-κB pathway in VS, which may be another potential mechanism. However, other studies demonstrated that there is no growth inhibitory effect for celecoxib on NF2-related VS or aspirin on sporadic VS [66,67]. Other studies have shown that NSAIDs, glucocorticoids, and other immunosuppressive drugs could not alter the expression of COX-2 in sporadic Sch [68].

### 7.2. Hsp90

Heat shock protein 90 (HSP90) is a ubiquitous molecule. The absence of Hsp90 results in proteasomal degradation [69]. The dysregulation of the Hippo pathway is necessary for schwannomagenesis, and MAPK signaling acts as a modifier for Sch formation. Furthermore, the pharmacological co-inhibition of YAP/TAZ transcriptional activity and MAPK signaling shows a synergistic size reduction in a mouse Sch model [70].

In a recent study, a novel small-molecule inhibitor compound of HSP90, NXD30001 (pochoxime A), was able to show reduced growth of NF2-deficient tumors in vivo. There are no current clinical trials using an HSP90 inhibitor [71].

The molecular patterns and mutations described for VS are summarized in Table 1.

## 8. Drug Repositioning

Mifepristone (RU486), a progesterone and glucocorticoid receptor antagonist that has already been approved for medical abortion, was chosen as the most promising candidate drug [72]. In a preclinical study, mifepristone reduced cellular proliferation in primary human VS cultures regardless of NF2 mutation. A phase II clinical trial on mifepristone in VS is currently being planned [72].

In VS, genes associated with NLRP3 were significantly upregulated in patients with poor hearing. NLRP3 mutation is associated with cochlear autoinflammation in conjunction with DFNA34-mediated hearing loss and age-rated hearing loss. The activation of NLRP3 triggers the production of IL-1β [73]. A recombinant human IL-1 receptor antagonist reversed the hearing loss observed in a family with sensorineural hearing loss and NLRP3 mutations [54].

## 9. Gene Therapy

Gene therapy offers the potential to treat a wide range of inherited and acquired human diseases. The direct modulation of affected genes in specific cell types represents the most powerful treatment strategy for NF2 patients. Delivery platforms typically include viral vectors, such as retroviruses, adenoviruses, and adeno-associated viruses (AAVs), as well as nonviral vectors, including nanoparticles and polymers [74].

A direct injection of an AAV serotype 1 vector encoding caspase-1 (ICE) under the Schwann-cell specific promoter led to the regression of Sch in a mouse model. Recently, a direct injection of AAV1 encoding the apoptosis-associated speck-like protein reduced tumor growth and resolved tumor-associated pain in a human xenograft Sch model [75].

Nonviral vectors, such as liposomal-, polymeric-, and peptide-based nanoparticles, offer an attractive alternative for gene delivery. Liposomes were used to deliver genome-editing agents to the cochlea of neonatal mice with dominant genetic deafness. By decorating the nanoparticle surface with a peptide targeting Schwann cells, peptide-based nanoparticles were used to deliver genetic materials, resulting in a decreased secretion of an ototoxic inflammatory cytokine from tumor cells [76].

## 10. Ongoing Clinical Trials

Table 2 shows ongoing clinical trials using multimodal treatment strategies for Sch. The superselective intraarterial infusion of bevacizumab is performed to control tumor progression (NCT01083966). Because of the promising results found with bevacizumab, it may be safely used by direct intracranial superselective intraarterial infusion up to a dose of 10mg/kg in order to enhance survival and hearing function. Another six trials are using medical treatment strategies. Crizotinib, AR-42 (OSU-HDAC42), everolimus, selumetinib (MEK 1/2 inhibitor), and tanezumab (a monoclonal antibody against nerve growth factor as a treatment for pain) are being evaluated in the trials. A previous meta-analysis suggests that there is insufficient evidence to recommend aspirin usage in patients with VS [77,78]. High-quality trials are warranted to determine the efficacy of aspirin in reducing VS growth (NCT03079999).

## 11. Future Direction

Bevacizumab has recently been considered as the first-line medical therapy for rapidly growing VS. Furthermore, new therapeutic strategies targeting the SH3PXD2A-HTRA1 fusion gene, several protein kinases, and the tumor microenvironment may be supportive for the mainstays of management. An immunotherapeutic approach may also be needed to control multiple tumor progression in the long term. In addition to the standard treatment strategy, including surgery and radiotherapy, these targeted medical therapies are needed for multiple and large tumors of VS (Figure 1). Multimodal therapy is required to manage patients with refractory VS.

The mainstays of management are observation, surgery, and radiation therapy. Bevacizumab has recently been considered as the first-line medical therapy for rapidly growing vestibular schwannomas. Furthermore, new therapeutic strategies targeting the SH3PXD2A-HTRA1 fusion gene, several protein kinases, and the tumor microenvironment may be supportive for the mainstays of management.

## Figures and Tables

**Figure 1 ijms-23-05462-f001:**
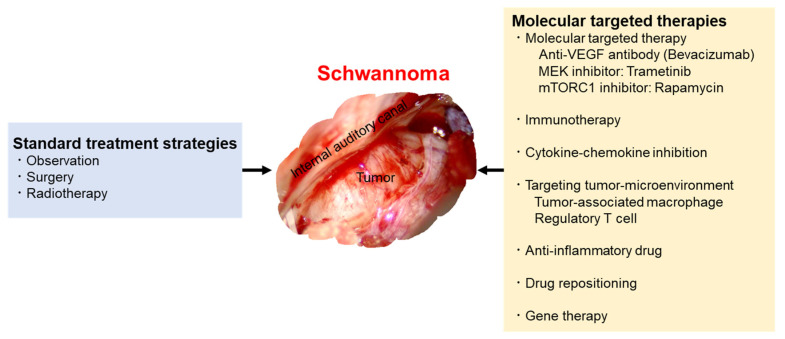
Multimodal treatment and management strategies.

**Table 1 ijms-23-05462-t001:** Molecular patterns and mutations currently described for VS.

Targeted Pathway	
**NF2 (merlin)-related pathway**	
1	Ras/Raf/MEK/ERK signaling
2	PI3K/Akt/mTORC1 signaling
**SH3PXD2A-HTRA1-fusion-related pathway**	
1	MAPK signaling
**Protein-kinase-related pathway**	
1	VEGF-A/VEGFR signaling
2	ErbB family signaling
3	PDGF/PDGFR signaling
4	HGF/HGFR (c-MET) signaling
**Cytokines and chemokines**	
1	CXCL12/CXCR4 signaling
2	IL-1β, IL-6, IL-34, M-CSF, TNF-α
**Tumor microenvironment**	
1	Tumor-associated macrophage
2	Regulatory T cell
3	PD-1/PD-L1
4	Hypoxia
**Inflammation and stress reaction**	
1	COX2
2	Hsp90

c-MET, c-mesenchymal–epithelial transition; COX2, cyclooxygenase 2; CXCL12, C-X-C motif chemokine ligand 12; CXCR4, C-X-C chemokine receptor type 4; ERK, extracellular-signal-regulated kinases; HGFR, hepatocyte growth factor receptor; Hsp90, heat shock protein 90; IL, interleukin; MAPK, mitogen-activated protein kinase; M-CSF, macrophage colony-stimulating factor; MEK, mitogen extracellular signal-regulated kinase; mTORC1, mammalian target of rapamycin complex 1; NF, neurofibromatosis; PDGFR, platelet-derived growth factor; PD-1, programmed death-1; PD-L1, programmed death-ligand 1; PI3K, phosphoinositide 3-kinase; Raf, rapidly accelerated fibrosarcoma; TNF-α, tumor necrosis factor-α; VEGF, vascular endothelial growth factor; VEGFR, vascular endothelial growth factor receptor.

**Table 2 ijms-23-05462-t002:** Active and recruiting clinical trials using medical therapeutic approaches for schwannoma.

ClinicalTrials.Gov Identifier	ID	RP	EE	Age	TS
NCT01083966	8, 2011	Lenox Hill Brain Tumor Center	30	≥18	Superselective intraarterial intracranial infusion of bevacizumab
NCT04283669	2, 2020	University of Alabama at Birmingham	19	≥6	Crizotinib
NCT03079999	6, 2018	Massachusetts Eye and Ear Infirmary	300	≥12	Aspirin
NCT02282917	9, 2015	Massachusetts Eye and Ear	5	≥18	AR-42 (OSU-HDAC42)
NCT01345136	7, 2015	University of California	4	16–65	Everolimus
NCT03095248	5, 2017	Children’s Hospital Medical Center	34	3–45	Selumetinib
NCT04163419	4, 2020	Massachusetts General Hospital	46	≥18	Tanezumab

ER, estimated enrollment; ID, initiation date; RP, responsible party; TS, treatment strategy.

## Data Availability

Not applicable.

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
