# Peer review of "A Critical Overview of Targeted Therapies for Vestibular Schwannoma"

_ijms, 2022, doi:10.3390/ijms23105462_

Round 1

Reviewer 1 Report

This paper is interesting and would deserve publication for the exhaustive overview it carries on the most recent molecular therapies.

On the other hand, it shows some areas lacking  knowledge of the issue, particularly when it deals with  surgery and radiotherapy. Overall speaking, lots of the conclusions of the paper derive from   uncompletely and inappropriate cited literature, where debated issues like surgery in vestibular schwannoma are, for instance, simplified in statements which  put at discussion the whole reliability of the work.

Within this premises, the paper is to be largely improved and major revision is required.

In details:

- table 1, page 7-8. why is trial NCT04801953 reported within the list of medical trials? Intraoperative nimodipine is NOT  a treatment for VS but an intraoperative surgical tool for nerves protection. Other papers support its use but within this context it is completely out of theme.

-Figure 1 is also out of theme. Why “intraoperative monitoring of facial nerve and auditory nerve” is cited in surgery? What is its meaning here? Why ABI? As such, these citations are inappropriate and do not belong   to the goals of the paper. I’d rather suggest to eliminate any reference to surgery because, as such, it evidences lack of knowledge and put at discussion the reliability of the whole work. Since this paper does not deal, or should not deal with the  different options of therapy of vestibular schwannoma, I would also recommend to eliminate any citation of radiosurgery (cyber-, gamma-knife ) as well as observation. This for the same reasons as explained before. The critical overview is to be limited to the targeted MEDICAL therapies. Surgery,  radiotherapy  and observation are not to be considered because the way they are cited and dealt with  are not along the current trends, it is better not to consider them and remain within the frame of the medical therapy.

-page 2, 4. protein Kinase -related pathway Lines 86, 87,.. when the authors write about Bevacizumab. Their conclusions on efficacy (tumor shrinkage, hearing preservation) should take into account the short follow up of the majority of VS bevacizumab series, and this aspect should be clarified, at least not ignored.

-page 6, 10. Ongoing clinical trials.

This paragraph is counfounding, does not add much to the topic of the paper, and mix different kinds of trials on medical therapy, surgical strategy, radiotherapy. Again, I strongly recommend to limit the report to the medical therapy

-page 8, 11. Future directions

Again, this paragraph mix considerations, non-updated evaluations of surgery and radiotherapy, and drive conclusion in too a fast manner. Digital tractography, which is also interesting as a preoperative tool too predict position of the facial nerve, is completely out of theme to be debated here. The conclusions on the best therapy are also too poorly supported not only by the paper, but by literature in general and it is not convenient to report such comments out of the theme.

The content of the present paper is an analysis of the current medical targeted therapy, and this has to remain the main aim of this paper.

Author Response

We are very grateful to the reviewers for their insightful comments and suggestions, which have undoubtedly helped us to improve our manuscript immensely. As indicated in the responses below, we have taken all their comments and suggestions into account when generating the revised version of the manuscript. Responses to the reviewers’ comments appear after the arrows, in blue text.

Reviewer 1:

This paper is interesting and would deserve publication for the exhaustive overview it carries on the most recent molecular therapies. On the other hand, it shows some areas lacking knowledge of the issue, particularly when it deals with surgery and radiotherapy. Overall speaking, lots of the conclusions of the paper derive from uncompletely and inappropriate cited literature, where debated issues like surgery in vestibular schwannoma are, for instance, simplified in statements which  put at discussion the whole reliability of the work.

Within this premises, the paper is to be largely improved and major revision is required.

Thank you very much for your review.

According to the reviewer’s comments, we have modified the discussion in the revised manuscript.

In details:

- table 1, page 7-8. why is trial NCT04801953 reported within the list of medical trials? Intraoperative nimodipine is NOT  a treatment for VS but an intraoperative surgical tool for nerves protection. Other papers support its use but within this context it is completely out of theme.

Thank you very much for your comments.

As the reviewer indicated, we have deleted the surgical and radiotherapeutic trials including NCT04801953.

-Figure 1 is also out of theme. Why “intraoperative monitoring of facial nerve and auditory nerve” is cited in surgery? What is its meaning here? Why ABI? As such, these citations are inappropriate and do not belong   to the goals of the paper. I’d rather suggest to eliminate any reference to surgery because, as such, it evidences lack of knowledge and put at discussion the reliability of the whole work. Since this paper does not deal, or should not deal with the  different options of therapy of vestibular schwannoma, I would also recommend to eliminate any citation of radiosurgery (cyber-, gamma-knife ) as well as observation. This for the same reasons as explained before. The critical overview is to be limited to the targeted MEDICAL therapies. Surgery,  radiotherapy  and observation are not to be considered because the way they are cited and dealt with  are not along the current trends, it is better not to consider them and remain within the frame of the medical therapy.

Thank you very much for your comments.

According to the reviewer’s comments, we have deleted the sentences about different options of therapy in the revised manuscript. Figure 1 has demonstrated the importance of these targeted medical therapies for multiple and large tumors of VS (in addition to the standard treatment strategy including surgery and radiotherapy). We have simplified Figure 1.

-page 2, 4. protein Kinase -related pathway Lines 86, 87,.. when the authors write about Bevacizumab. Their conclusions on efficacy (tumor shrinkage, hearing preservation) should take into account the short follow up of the majority of VS bevacizumab series, and this aspect should be clarified, at least not ignored.

Thank you for your comments, we have added the indicated aspect of the short follow up of the majority of VS bevacizumab series in the revised manuscript.

-page 6, 10. Ongoing clinical trials.

This paragraph is counfounding, does not add much to the topic of the paper, and mix different kinds of trials on medical therapy, surgical strategy, radiotherapy. Again, I strongly recommend to limit the report to the medical therapy

According to the reviewer’s comments, we have deleted the trials about different options of therapy in the revised manuscript.

-page 8, 11. Future directions

Again, this paragraph mix considerations, non-updated evaluations of surgery and radiotherapy, and drive conclusion in too a fast manner. Digital tractography, which is also interesting as a preoperative tool too predict position of the facial nerve, is completely out of theme to be debated here. The conclusions on the best therapy are also too poorly supported not only by the paper, but by literature in general and it is not convenient to report such comments out of the theme.

According to the reviewer’s comments, we have modified the section of future directions in the revised manuscript.

-The content of the present paper is an analysis of the current medical targeted therapy, and this has to remain the main aim of this paper.

Thank you very much for your comments. We have focused on the current medical targeted therapy, as the reviewer indicated.

Reviewer 2 Report

I congratulate the authors for their impressive work in comprehensively reviewing the current literature on the molecular biology of VS and related therapeutic approaches.

Except for some minor typos found across the manuscript, I feel that the paper can be accepted in its current form.

As a side note, I suggest the authors create a separate Table where they summarize all molecular patterns and mutations currently described for VS. This would be very helpful for improving the easy detection of molecular patterns for all readers.

Author Response

We are very grateful to the reviewers for their insightful comments and suggestions, which have undoubtedly helped us to improve our manuscript immensely. As indicated in the responses below, we have taken all their comments and suggestions into account when generating the revised version of the manuscript. Responses to the reviewers’ comments appear after the arrows, in blue text.

Reviewer 2:

I congratulate the authors for their impressive work in comprehensively reviewing the current literature on the molecular biology of VS and related therapeutic approaches.

Except for some minor typos found across the manuscript, I feel that the paper can be accepted in its current form.

As a side note, I suggest the authors create a separate Table where they summarize all molecular patterns and mutations currently described for VS. This would be very helpful for improving the easy detection of molecular patterns for all readers.

Thank you very much for your review.

As the reviewer indicated, we have corrected a misspelled word in the revised manuscript. A separate Table summarizing molecular characteristics was made.